# Cloud-Based Image Retrieval Using GPU Platforms

**Sidi Ahmed Mahmoudi [1],\*** , **Mohammed Amin Belarbi [1]**, **El Wardani Dadi [2]** ,
**Saïd Mahmoudi [1]** and **Mohammed Benjelloun [1]**

[1] Department of Computer Science, Faculty of Engineering, University of Mons, 7000 Mons, Belgium;
    MohammedAmin.BELARBI@umons.ac.be (M.A.B.); said.mahmoudi@umons.ac.be (S.M.);
    Mohammed.benjelloun@umons.ac.be (M.B.)
[2] LaRi Laboratory, National School of Applied Sciences, Al Hoceima, University of Mohammed First,
    Oujda 60000, Morocco; wrd.dadi@gmail.com
\* Correspondence: sidi.mahmoudi@umons.ac.be

**Abstract:** The process of image retrieval presents an interesting tool for different domains related to computer vision such as multimedia retrieval, pattern recognition, medical imaging, video surveillance and movements analysis. Visual characteristics of images such as color, texture and shape are used to identify the content of images. However, the retrieving process becomes very challenging due to the hard management of large databases in terms of storage, computation complexity, temporal performance and similarity representation. In this paper, we propose a cloud-based platform in which we integrate several features extraction algorithms used for content-based image retrieval (CBIR) systems. Moreover, we propose an efficient combination of SIFT and SURF descriptors that allowed to extract and match image features and hence improve the process of image retrieval. The proposed algorithms have been implemented on the CPU and also adapted to fully exploit the power of GPUs. Our platform is presented with a responsive web solution that offers for users the possibility to exploit, test and evaluate image retrieval methods. The platform offers to users a simple-to-use access for different algorithms such as SIFT, SURF descriptors without the need to setup the environment or install anything while spending minimal efforts on preprocessing and configuring. On the other hand, our cloud-based CPU and GPU implementations are scalable, which means that they can be used even with large database of multimedia documents. The obtained results showed: 1. Precision improvement in terms of recall and precision; 2. Performance improvement in terms of computation time as a result of exploiting GPUs in parallel; 3. Reduction of energy consumption.

**Keywords:** image retrieval; SIFT; SURF; cloud computing; GPU computing

## 1. Introduction

Multimedia content-based retrieval in large databases is an active topic in various research communities such as video surveillance, 3D models analysis, plant leaf retrieval [1], computer aided diagnosis (CAD) and pattern recognition. Indeed, large databases of multimedia data (2D images, videos, 3D objects) became more and more available recently. However, when the dataset size gets very large, the retrieving process becomes very challenging due to the hard management of storage, computation speed and similarity representation. In the literature, several methods and algorithms can be applied for content-based image retrieval systems such as invariant features (SIFT [2] and SURF [3]), points of interest, contours and mean projection transform. These algorithms can also be applied for content-based image retrieval (CBIR), content-based video retrieval (CBVR), and content-based storage retrieval (CBSR). In the same way, similarity measurements can be used for this kind of systems. In order to accelerate the retrieval process and achieve large-scale retrieval, various content-based methods and approaches have been proposed in the literature [4].

The latter exploit the high-performance computing of different processors such as multi-core, GPU, and multi-GPU, etc. Despite the high efficiency of these solutions, the problem is that most of them are partial and do not cover the whole retrieving process. In this context, several computer vision algorithms and applications tend to benefit from the high computing power of multi-CPU or/and multi-GPU platforms by the development of parallel solutions. Notice that image and video processing algorithms, and more particularly SIFT and SURF [3] descriptors are well adapted for parallelization within multi-CPU or/and multi-GPU platforms since they consist mainly of a common computation over many pixels [5,6]. Several parallel solutions that exploit the above-mentioned hardware have been developed recently. Although they offer a great potential of processors (multi-CPU or/and multi-GPU, cluster, grid, etc.), the use, configuration and exploitation of these solutions in not so easy. Indeed, users must have the required hardware and need to download, install and configure the related CPU or/and GPU libraries.

Therefore, we propose a cloud-based platform that groups and integrates image and video processing algorithms, which are exploited and combined for providing an efficient method of image indexation and retrieval. The proposed combination of descriptors is well adapted for dimensionality reduction, where the selection of the most significant values of descriptors (using PCA method) allowed to reduce the research time with the maintain of precision [7]. As a result, our method is well suited for large scale image retrieval.

For the platform, each connected guest or user (to our platform) can select the required application, load its data and retrieve results with an environment similar to desktop either if the required application exploits parallel (GPU) or heterogeneous (multi-CPU/multi-GPU) platforms. Our cloud-based image retrieval method can be executed in real time and in a secure way. The related libraries and hardware drivers are automatically integrated and configured in order to offer to users an access to the different algorithms without the need to download, install and configure software and hardware. Moreover, the platform offers the access to the integrated application from multiple users thanks to the use of docker [8] containers and images.

The remainder of the paper is organized as follows: Section 2 presents the related works. In Section 3, we present our GPU-based and hybrid method of image retrieval that combines SIFT and SURF descriptors. Section 4 describes our related cloud-based solution for image retrieval. In Section 5, experimental results are presented and discussed. Finally, conclusions and future works are discussed in the last Section.

## 2. Related Work

In literature, we can cite several works related to the domain of cloud-based image retrieval. These works can be presented within two subsections: (1) content-based image retrieval systems. (2) Cloud-based computer vision platforms.

### 2.1. Content-Based Image Retrieval Systems

In this paper we are focused on content-based image retrieval (CBIR) systems, which can be exploited for 3D objects and videos research and indexation since both of them (videos and 3D objects) are composed from 2D images [9]. Theses retrieval systems are generally based on the same operating philosophy that, given a query document, retrieve similar documents in the database. The process of retrieval is performed in two essentials phases: indexing phase and matching phase.

- Indexing phase: this phase consists of designing an efficient canonical characterization of the multimedia document. This characterization is referred to as a descriptor or a signature, it serves as a key in the search process. The principal step of this phase is the features extracting. The indexing phase is the principal problematic of content-based image retrieval that scientific committee is working on. Indeed, designing an efficient canonical characterization of a given multimedia document still remains a major challenge, it is a critical kernel with a strong influence on the

retrieval performances (i.e., computational efficiency and relevance of the results). The bulk of that challenge lies in the step of features extracting, which is the principal step of the indexing phase. In order to design an efficient method of features extraction, several algorithms have been proposed in the literature, [2,3,10]. Otherwise, several solutions are proposed for feature extraction using deep neural networks that consist of learning from annotated data before applying inference within the generated model [11,12]. Notice that the deep learning approach provides high precision due to the generation of very large sets of features but it requires high intensive computation and already annotated data. In this context, some GPU implementations are proposed in [13,14].

- Matching phase: this phase consists of comparing the descriptor of the query with the descriptor of each multimedia document in the database. This comparison is performed using a dissimilarity measure, that computes the distances between pairs of descriptors. The existing Multimedia content-based retrieval methods use different well known dissimilarity measurements like, K-nearest neighbor (KNN), Euclidean L2, Minkowski, KLD, etc. The reader can refer to the survey of [15] to get more details about these similarity measures. Authors in [16] proposed a content-based synthetic aperture radar (SAR) image retrieval approach for searching SAR image patches. This method is based on a similarity measure named region-based fuzzy matching (RFM) and relevance feedback for improving precision. Otherwise, one can find methods that use 3D specific measurements like the CM-BOF method [17], used for 3D shape retrieval. The latter is based on a measurement function called clock matching. In order to accelerate the matching process, several techniques based on distributed computing have been proposed in [18]. These solutions use CPUs only, while others implementations are implemented in parallel with GPU platforms [19,20]. Authors in [21] proposed a shape retrieval method using distributed databases that allowed to increase result precision.

## 2.2. Cloud-Based Computer Vision Platforms

Recent development of computer vision platforms have been significantly influenced by the emergence of a growing number and accessible cloud computing platforms hosted by large-scale IT-companies (AWS, GCP, Azure). They enabled the development of a variety of cloud interfaces, which makes abstraction on the complexity behind computer vision application. The latter use a specific workflow for cloud architectures [22], which gives access to a high computing power without the need of a low-level software programming or any hardware adaptation.

CloudCV [23] is an example of a cloud-based and distributed computer vision platform composed of three parts; an AI-as-a-service platform that enables researchers to easily convert their deep learning models to web service and call them by a simple API, a drag and drop collaborative platform for building models, an evaluation server for comparing different AI and computer vision algorithms (for example for challenges). CloudCV provides access to two API (Python and Matlab), and englobes multiple modern components for its backend architecture such as OpenCV, Caffe, Turi (GraphLab).

The image processing on line (IPOL) [24] platform provides image processing algorithms and descriptions, source code and a handy web interface to check results from new input images. This initiative intended to promote reproducible research related to image processing. Other cloud commercial applications such as Face API (by Microsoft Azure), Amazon Rekognition, Watson Visual Recognition (by IBM) or Clarifai, deliver APIs, which focus on specific computer vision tasks for image and video understanding.

On the other hand, with the exponential growth of image data, solving big data challenges becomes an important task managed on the cloud. Yan et al. [25] proposed a cloud architecture dedicated to large scale image processing based on Hadoop. They evaluated the performance of the platform using different image processing algorithms. They reported some issues with data distribution and cluster resource related to the use of Hadoop. Recently, we developed real time web-based toolbox for computer vision [26,27] that integrates several classic image processing algorithms.

In terms of security, authors in [28] explained that the best way to communicate between machines (or containers) is the use of secured protocols such as FTP [29], SSH [30], SFTP [31] and SCP. In [32], authors demonstrated that the protocol HTTPS actually presents the best solution for a web server, since it combines between HTTP protocol and an encrypted connection (ensured by the transport layer security, or its predecessor, secure sockets layer). In our case, we have chosen the famous protocol HTTPS for our web server, and SFTP protocol to ensure secure transfer of data within our platform. Notice that SFTP is provided with SSH protocol that allows to execute commands and run the applications. Our contribution can be summarized with three points:

1. The development of an efficient method of content based image retrieval that combines the descriptors of SIFT and SURF;
2. A portable GPU implementation that allows to accelerate the process of indexation and research within multimedia databases. This implementation allows to exploit both NIVIDIA and AMD/ATI cards;
3. Cloud-based implementation that allows an easier exploitation of our GPU-based method without the need to download, install and configure software and hardware. The platform handles multi user connection based on docker container orchestration architecture.

## 3. GPU-Based Hybrid Multimedia Retrieval

### 3.1. Sequential Solution

Before presenting the GPU-based implementation, we start by describing the main steps of our algorithm of image indexation and matching. The latter can be summarized within three main steps (Figure 1): pre-processing, indexation and research.

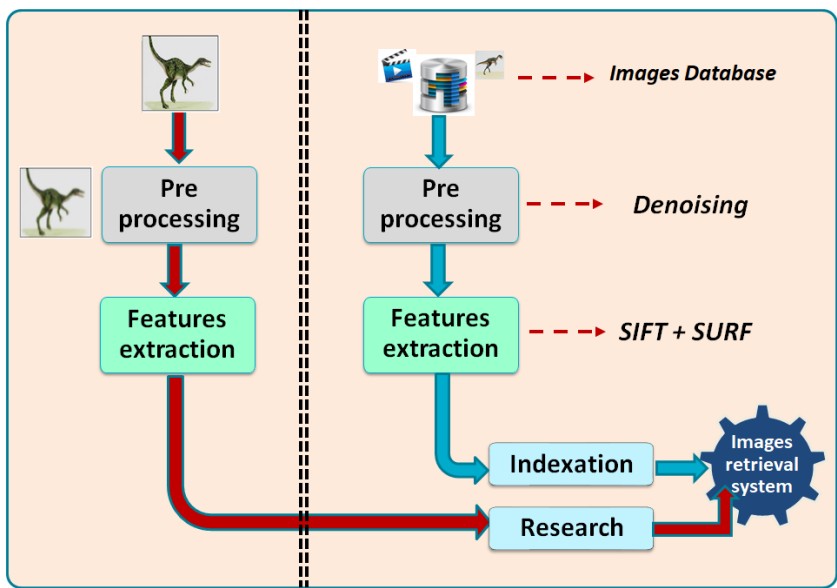

**Figure 1.** The process of image indexation and matching.

1. Pre-processing

   The methods of image retrieval and indexation are mainly based on features extraction algorithms that allow to detect the main key points of images. In order to improve this process, we apply a pre-processing step that allows to reduce noise and consequently the number of key points. For this aim, we have tested several image filters such as median, bilateral and Gaussian filters. In our case, the Gaussian filter provided the best results for reducing the number of key points. This reduction allowed to reduce the computation time of the next steps (features extraction).

2. Indexation

This step was performed offline and is so time-consuming since it is applied on the entire database. After the pre-processing step, we apply the feature extraction within SIFT and SURF descriptors. Indeed, we started by computing SIFT features for each image from the dataset, the result is represented by a matrix of (n × m) lines and 128 columns. Notice that $n$ and $m$ represent the weight and height of image. Secondly, we applied the same process using SURF descriptor, which requires less execution time but provides less precise results. The result of the SURF descriptor was represented by a matrix of (n × m) lines and 64 columns. Once the steps of SIFT and SURF were completed, we can combine their results (two matrices) with one matrix only. Since the two descriptors (SIFT and SURF) present a different number of columns, we have increased (using zero values) the size of SURF descriptor to 128 values in order to be compatible with the size of SIFT descriptor. Finally, we got one matrix with 2 × (n × m) lines and 128 columns. The latter was used for indexing the image database. As a result, we obtain three folders: SIFT, SURF and SIFT + SURF descriptors. This step was performed offline and was so consuming in time since it was applied on the entire database. After the pre-processing step, we applied the feature extraction within SIFT and SURF descriptors. Indeed, we started by computing SIFT features for each image from the dataset, the result is represented by a matrix of (n × m) lines and 128 columns. Notice that $n$ and $m$ represent the weight and height of image. Secondly, we applied the same process using SURF descriptor, which requires less execution time but provides less precise results. The result of SURF descriptor is represented by a matrix of (n × m) lines and 64 columns. Once the steps of SIFT and SURF were completed, we can combine their results (two matrices) with one matrix only. Since the two descriptors (SIFT and SURF) present a different number of columns, we have increased (using zero values) the size of SURF descriptor to 128 values in order to be compatible with the size of SIFT descriptor. Finally, we got one matrix with 2 × (n × m) lines and 128 columns. The latter is used for indexing image database. As a result, we obtained three folders: SIFT, SURF and SIFT + SURF descriptors.

3. Research

Unlike the previous step, this phase was performed online where the user can provide its query image and choose the preferred algorithm for features extraction (SIFT, SURF or both). Once the user choice was provided, the query image was smoothed (pre-processed) within a Gaussian filter and characterized within the previously selected algorithm. The next step consisted of comparing the query image features with those of image database. The comparison was performed within two similarity matching methods: Flann based matcher and Brute force matcher [7]. Our selection of these two methods was due to their efficiency and fast execution, which is so important in our case.

The final result of a similarity measurement was a normalized value, ranged between 0 and 1. The value of 0 represented complete similarity and the value of 1 represented complete dissimilarity. An algorithm of KNN [33] was used to retrieve the similar images. In our cases we showed the top 50 images similar to the query image.

*3.2. Parallel Solution*

Despite the high accuracy of the above-mentioned method, its computing time is so significant, which makes our method not adapted for image indexation and retrieval within large databases. The high computing time is due to:

- Several image processing algorithms applied for each image within the indexation phase.
- The use of high definition images that require more time for features extraction.
- The high computational intensity of features extraction and distance computation steps.

These steps have been identified within a profiling algorithm that measured the computation time and memory space of each step of our algorithm. The profiling result is approved by the application of our complexity estimation equation defined in [34,35].

To overcome this constraint, we developed a GPU-based portable implementation that can exploit both NVIDIA and ATI GPUs. For a better exploitation of GPUs, we ported the whole process of image indexation and retrieval on GPU, by implementing (in parallel) all the steps on GPU: pre-processing, SIFT and SURF descriptors and distance computation. This allowed us to accelerate the process of computation and reduce the data transfer times since there is no need to transfer intermediate data from CPU to GPU memory.

### 3.2.1. CUDA Implementation

The API CUDA is used for exploiting NVIDIA cards for image indexation and retrieval method. Our CUDA implementation is summarized by two steps:

1. CUDA-based image indexation: we used the GPU module of OpenCV library (OpenCV GPU Module, https://docs.opencv.org/2.4/modules/gpu/doc/introduction.html) for implementing the functions of pre-processing and SURF descriptor. The CUDA implementation of SIFT descriptor is provided from [36]. These GPU functions consist of applying the operations of features extraction in parallel using the same number of CUDA threads as the number of image pixels. With this, each CUDA thread can apply its treatment on one pixel value and all the CUDA threads are launched in parallel. Since the indexation phase requires the transfer of all images (database), we use the CUDA streaming technique in order to overlap image data transfers by CUDA functions execution.
2. CUDA-based image matching: the query image is also analyzed within the above-mentioned CUDA functions (pre-processing, SIFT and SURF descriptors). In addition to these functions, this step requires the computation of distance between the query image features and the database features. This distance, computed within FLANN-based matcher and Brute force matcher, is also implemented using the GPU module of OpenCV library.

### 3.2.2. OpenCL Implementation

The OpenCL framework is used for exploiting ATI/AMD graphic cards for our image indexation and retrieval method. Our OpenCL implementation is also summarized by two steps:

1. OpenCL-based image indexation: we used the OpenCL module of OpenCV library (OCL module, https://docs.opencv.org/2.4/modules/ocl/doc/introduction.html) for implementing the functions of pre-processing and SURF descriptor. The OpenCL based implementation of SIFT descriptor is provided from [37]. This implementation is so similar to the corresponding CUDA version. The main difference between CUDA and OpenCL methods is that with OpenCL, we have to create a context in order to specify the device. In this way, the same code can be used for programming either CPU or GPU. Notice that in this case, we do not overlap data transfers by execution since the streaming option is not provided in OpenCL.
2. OpenCL-based image retrieval: this step is also implemented using the OpenCL module of OpenCV library for extracting features (pre-processing, SIFT and SURF descriptors) of the query image. The distance is ported on OpenCL using the Brute force matcher.

The above-mentioned CUDA and OpenCL implementations are used to provide a portable GPU-based method of image retrieval and indexation. Indeed, the program starts by detecting the type of available GPU. In case of NVIDIA cards, CUDA implementations are called. Otherwise, the OpenCL implementations are called for exploiting ATI graphic cards. In this case, an OpenCL context is created for specifying the GPU for computation.

### 4. Cloud-Based Hybrid Multimedia Retrieval

To ensure good performance of our cloud-based application, we have used a virtual machine (VM) that allows the access to our method of image indexation and research. As shown in Figure 2, the related web address is: https://www.multimedia-processing.com/mmr.php. Users are invited to test the application within this address. This application is developed using PHP (PHP, https://www.php.net/) and Bootstrap (Bootstrap, https://getbootstrap.com/) that allowed to have a multi-platform website running even on mobile devices (smartphone, tablet, etc.). The access to our cloud-based application is secured within HTTPS protocol.

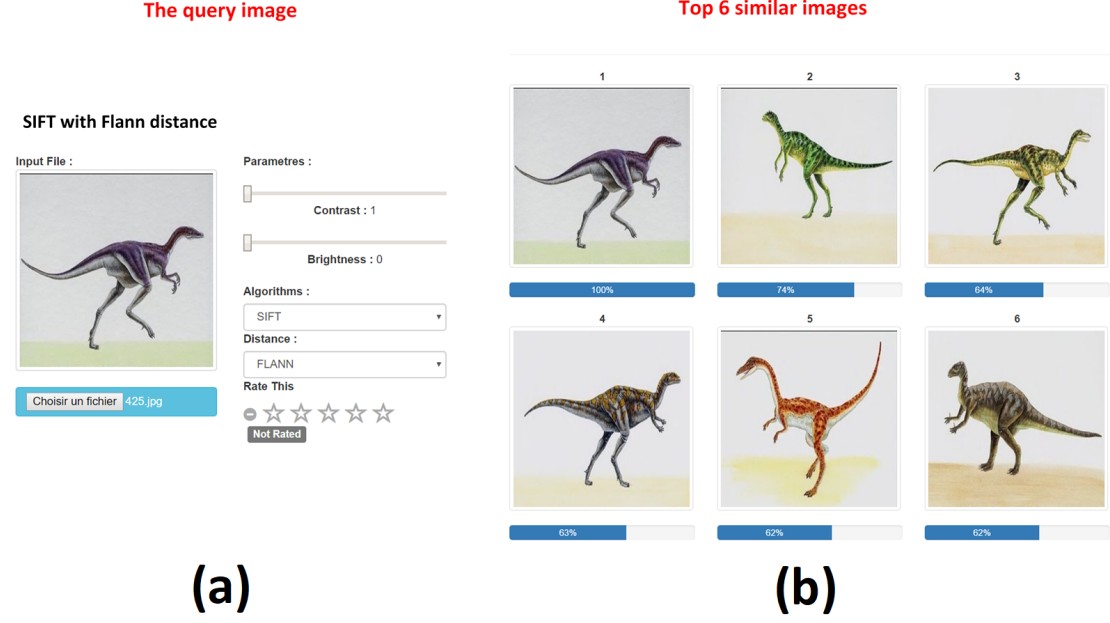

**Figure 2.** (**a**) Query image and algorithm selection. (**b**) Cloud-based multimedia retrieval result (top 6).

Otherwise, we used the docker framework in order to provide a multi-user exploitation where different users can run the same application simultaneously (Figure 3). Docker framework allows to deploy the applications without the need to install operating systems. Notice that docker is an open source platform released in 2013 and used for the creation, deployment and management of applications. Docker is mainly based on images and containers where images allow us to define the precise software packages (applications, libraries, configurations, etc.). Images can be also created by combining or modifying other standard images downloaded from public repositories. On the other hand, containers present instances of images that can be executed from each user (one user can execute one container). Docker containers are isolated and are run on single operating systems which makes them so lightweight than virtual machines. To summarize, docker containers present an open source software platform of development. Its main advantage is the ability to package applications in containers, which allows them to be portable among any system running the Linux operating system (OS). With docker, we generated and configured an image including the operating system (Ubuntu) and the required library (OpenCV) for image feature extraction. This image is called "basic-docker-image" in our case. Then, we generated a second image "nvidia-docker" that allows to exploit NVIDIA GPUs within the GPU module of OpenCV and CUDA. Finally, we have generated a third image "opencl-docker" that allows us to exploit ATI/AMD GPUs within the OpenCL module of OpenCV and OpenCL. The process of exploiting our cloud application is summarized within four steps:

- Web selection: the user selects the application of image retrieval within the platform.
- Input parameters uploading: the user provides the input parameters (the query image, the type of algorithm and preferred hardware) that will be sent to the web server.
- Cloud-based execution: at this moment, the cloud platform generates the related docker container (basic, nividia or opencl) with all the parameters in order to execute the application. Notice that in case we have many users, the platform creates a container for each user.
- Results presentation: at the end of the process of research, all the containers will be removed by the cloud platform and show the results to the user. Figure 4 illustrates the general architecture of our cloud platform.

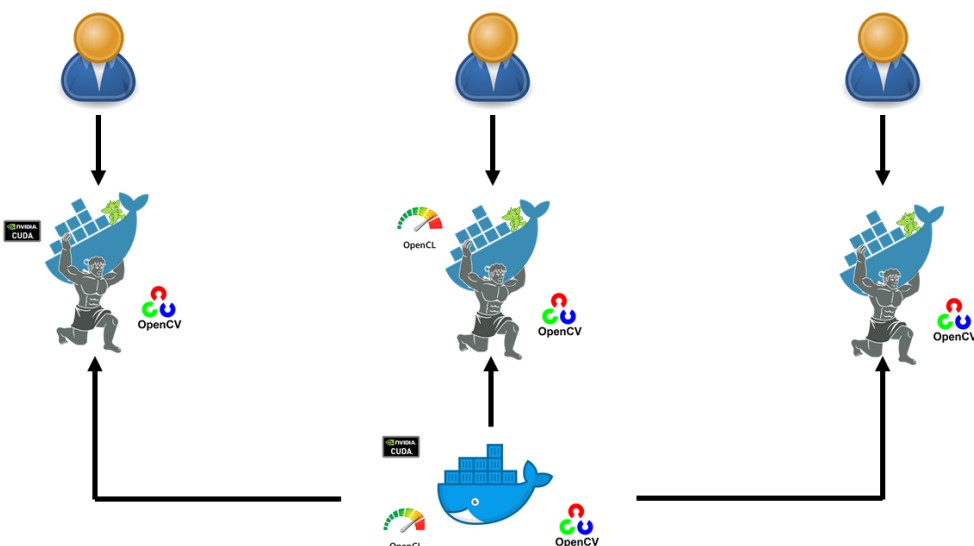

**Figure 3.** Multi-user execution.

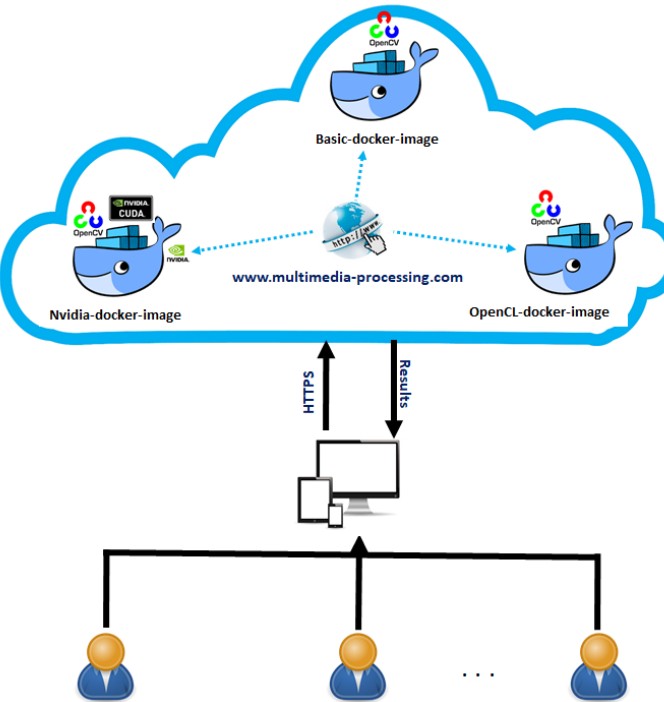

**Figure 4.** The cloud-based platform.

## 5. Experimental Results

The tests were run on the following local hardware:

- CPU: Intel Core (TM) i5, 2520M CPU@ 2.50 GHz, RAM: 4 GB;
- GPU NVIDIA: GeForce GTX 580, RAM: 1.5 GB, 512 CUDA cores.

The cloud-based implementation was executed using two virtual machines provided from the Google cloud platform:

- CPU: Intel(R) Xeon(R) CPU E5-2650L v4 @ 1.70 GHz, RAM: 1 GB
- GPU: 4 GPU Nvidia GTX 980, RAM: 4 GB

Experimentations have been conducted using three image databases:

1. Wang databse (Wang database, http://wang.ist.psu.edu/docs/related/): 10,000 low resolution images of size 128 × 85 classed in 100 categories, where each class contains 100 images;
2. Corel-10k database (Corel-10k database, http://www.ci.gxnu.edu.cn/cbir/Dataset.aspx): 10,000 images of size 192 × 128, classed in 100 categories, where each class contains 100 images;
3. GHIM-10k database (GHIM-10k database, http://www.ci.gxnu.edu.cn/cbir/Dataset.aspx): 10,000 images of size 300 × 400, classed in 20 classes where each category contains 500 images.

In terms of accuracy, the proposed method of image indexation and research provided an accurate result. As shown in Figure 5, our method outperformed the state of the art algorithms, within the three databases, in terms of recall and precision (R/P) when selecting the most similar 50 images (top 50). This high accuracy was due to the combination of SIFT and SURF descriptors that allowed to detect more features. Moreover, the pre-processing step allowed to improve the quality of detected features. We note more accurate results with the Wang database since it presents low resolution images. We plan in the future to improve the obtained accuracy by combining these features with deep learning features.

Otherwise, our CUDA and OpenCL implementations allowed us to provide an accelerated method which can exploit both NVIDIA and ATI graphic cards. This acceleration allowed us to reduce both indexation and research phases, as shown in Table 1. Notice that the use of GPU offered low acceleration in case of processing low resolution videos. This was due to the weak exploitation of graphic processing units. Indeed, GPUs were more adapted for massively parallel applications. We note also that CUDA offered better performance than OpenCL since CUDA presented the most performant GPU programming language. Moreover, unlike CUDA, OpenCL did not offer the possibility to overlap data transfers by kernels executions. Notice that the OpenCL performance were obtained using a GPU NVIDIA. This allowed us to obtain a fair comparison of performance. Our OpenCL implementation was developed in order to offer a portable solution for image indexation and research.

Finally, the cloud-based implementation offered the same accuracy since the same algorithm is applied on a cloud platform. The performance was slightly reduced, which was due to the transfer times between local and cloud machines. Notice that GPU version (in cloud) provides also better performance than CPU and mainly when using CUDA. This cloud version allowed us to provide a solution for users without the need to download, install and configure the related hardware and software.

**Table 1.** GPU performances and acceleration of image retrieval steps (image of 1920 × 1080 pixels).

| Algorithm | 2 CPU | GPU (CUDA) | | GPU (OpenCL) | |
|---|---|---|---|---|---|
| | | Time | Acc (x) | Time | Acc (x) |
| Pre-processing | 0.24 s | 0.002 s | 120× | 0.003 s | 80× |
| SURF descriptor | 0.54 s | 0.120 s | 4.50× | 0.15 s | 3.60× |
| SIFT descriptor | 0.69 s | 0.130 s | 5.31× | 0.16 s | 4.31× |

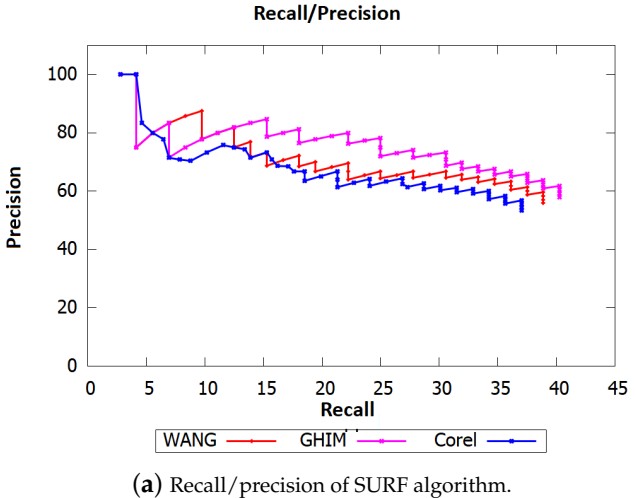

(**a**) Recall/precision of SURF algorithm.

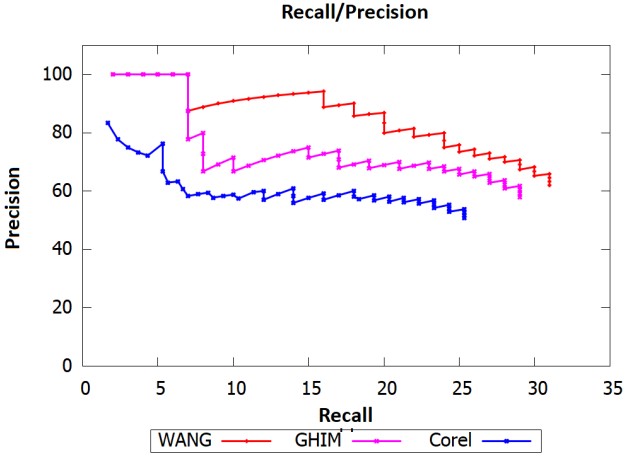

(**b**) Recall/precision of SIFT algorithm.

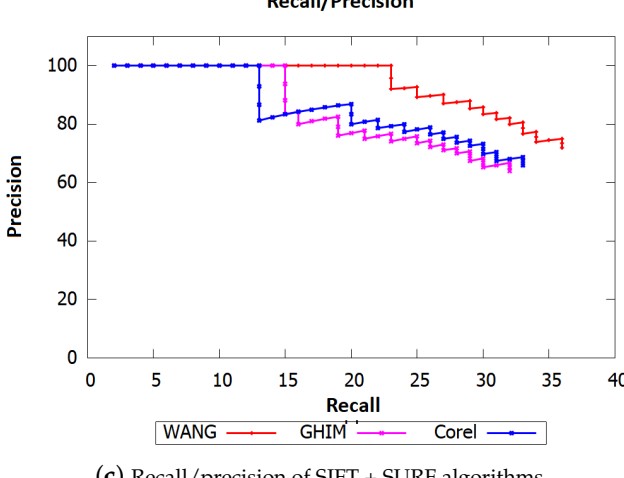

(**c**) Recall/precision of SIFT + SURF algorithms.

**Figure 5.** Recall/Precision of SIFT and SURF algorithms for the top 50.

## 6. Conclusions

In this paper, we have presented a cloud-based application of image indexation and matching that allows to exploit both nvidia and ATI graphic cards. The proposed application offers three

main benefits: (1) efficient image retrieval thanks to the combination of SIFT and SURF descriptors; (2) simple, easy and multi-user exploitation within the cloud platform; (3) fast execution as a result of the parallel implementation exploiting nividia and ATI graphic cards. Experimental results demonstrated the efficiency of our solution in terms of recall/precision, computation time and multi-user exploitation. As future works, we plan to exploit deep learning methods in order to improve that quality of training, indexation and matching. We plan also to apply the same process for content-based video retrieval.

**Author Contributions:** Conceptualization, S.A.M.; Formal analysis, M.A.B.; Investigation, E.W.D.; Methodology, S.A.M.; Project administration, S.A.M.; Software, S.A.M.; Supervision, S.M. and M.B.; Validation, S.A.M. and M.A.B.; Writing—original draft, E.W.D.; Writing—review & editing, S.A.M.

**Funding:** This research received no external funding.

**Conflicts of Interest:** The authors declare no conflict of interest.

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
