# Peer review of "Cloud-Based Image Retrieval Using GPU Platforms"

_computers, doi:10.3390/computers8020048_

Round 1
Reviewer 1 Report
This paper introduces a cloud-based platform for content-based image retrieval. The manuscript is written and organized well. However, I’d like to suggest the authors to enrich the experiments. More image databases could be considered to verify their method. Also, are there any comparisons could be added to prove the effectiveness of your approach? Furthermore, the following literature could be added to enrich your reference.
Metric-Learning based Deep Hashing Network for Content Based Retrieval of Remote Sensing Images
SAR image content retrieval based on fuzzy similarity and relevance feedback
Unsupervised deep feature learning for remote sensing image retrieval
Author Response
Paper: Cloud-based Images Retrieval using GPU Platforms
Responses to reviewers
1. Question 1: this paper introduces a cloud-based platform for content-based image retrieval. The manuscript is written and organized well. However, I’d like to suggest the authors to enrich the experiments. More image databases could be considered to verify their method.
Response 1: thank for the interesting remarks and suggestions. We have enriched our experiments (Section 4) in the new version using larger and more diversified databases such as Corel-10k and GIHIM. The Corel database presents 10000 images of size 192x128, classed in 100 categories where each class contains 100 images. The GHIM-10k database contains also 10000 images of size 300x400, classed in 20 classes where each category contains 500 images. These experiments confirmed our previous results by the improvement of accuracy thanks to the efficient combination of SIFT and SURF descriptors.
2. Question 2: also, are there any comparisons could be added to prove the effectiveness of your approach?
Response 2: Indeed, as consequence of increasing our databases, we measured the precision (Section 4) in terms of Recall/Precision when selecting more images (Top 50). Notice that in the previous version, the precision was computed for the most similar 20 images (Top 20).
3. Question 3: furthermore, the following literature could be added to enrich your reference.
- Metric-Learning based Deep Hashing Network for Content Based Retrieval of Remote Sensing Images.
- SAR image content retrieval based on fuzzy similarity and relevance feedback.
- Unsupervised deep feature learning for remote sensing image retrieval
Response 3: The proposed references are included and cited in the new version of the paper (Section 1.1). These references are well related to our multimedia retrieval approach. Actually, we work on exploiting unsupervised deep features for improving the precision of our approach when using massive volumes. The use of relevance feedback presents also one our main perspectives. This allows to exploit the user feedback in order provide more relevant results for queries.

Reviewer 2 Report
Cloud-based CPU and GPU implementation of image retrieval which offers users to access different methods such as SIFT and SURF. For this goal, a hybrid retrieval method by the combination of SIF and SURF is developed which claimed to be efficient with low consumed energy. This paper addressed an important problem of image retrieval which is the computational time, however, a revision is required. Please consider bellow comments.
The abstract is not well-written. The research gap and the solution are presented in the last part of the abstract. This needs to be discussed earlier. The challenges such as management of large dataset in terms of storage and temporal performance are mentioned without giving proper solution in the abstract. What do you mean by combining SIFT and SURF? This should be discussed in one sentence how you combined them. The sentence “show the efficiency of our solution in terms of recall/precision” is not accurate. There is no relation between efficiency and precision or recall. These metrics measure the accuracy or performance.
Please provide a proper reference for (Markel, 2014) in line 51. You mentioned about three different retrieval categories (2D images, video, and 3D objects). If your method works with all these data, you need to describe how your method can work with all these data in the introduction.
Text in Figure 1 and 2 hardly visible. You also may consider combining these figures by including the pre-processing step and images from figure 2 into figure 1. The title mentioned “Image Retrieval” but you present “Multimedia Retrieval” which are not consistent. CAD is not defined. Make sure define all abbreviations in the manuscript. Please provide ref for the usage of image retrieval in different communities in the early introduction, for example, consider these works, shape-based retrieval, and analysis of 3D models, and, invariant feature descriptor based on harmonic image transform for plant leaf retrieval. Please consider using other feature-based methods such as, contour-based corner detection and classification by using mean projection transform, in line 26.
Are these categories “1. Multimedia content-based retrieval systems. 2. Cloud-based computer vision platforms” are available in the literature or this is your finding? If you categorized this, you need to discuss why, if it is available in the literature, please include a proper citation. Please consider discussing matching methods such as, invariant feature matching for image registration application based on new dissimilarity of spatial features, and, a region based fuzzy feature matching approach to content-based image retrieval, work in matching phase section.
Why image features in figure 1 shows like a database? Moreover, the flow from the input image is from the bottom to the top, it is recommended to start from the top. Text in Figure 3,4, and 4 hardly visible. Please include a reference (in the figure caption) to the image dataset used in Figure 3. A combination of SIFT and SURF mentioned in the abstract is not discussed properly in the method section.
You’ve got 100% precision and recall based on Figure 6(c) which is not realistic. Probably not enough data is used or you used a simple retrieval dataset. Moreover, it seems Wang dataset only contains 2D images while you mentioned about video and 3D objects earlier which is not consistent. You need to evaluate video and 3D objects, or alternatively, just talk about 2D image retrieval in the entire manuscript including the title.
How many images did you use to produce Figure 6 results? Have you used all 10,000 + 1000 image in Wang dataset?
Author Response
a. Reviewer 2:
1. Question 1: Cloud-based CPU and GPU implementation of image retrieval which offers users to access different methods such as SIFT and SURF. For this goal, a hybrid retrieval method by the combination of SIF and SURF is developed which claimed to be efficient with low consumed energy. This paper addressed an important problem of image retrieval which is the computational time, however, a revision is required. Please consider bellow comments. The abstract is not well-written. The research gap and the solution are presented in the last part of the abstract. This needs to be discussed earlier.
Response 1: thank for the interesting remarks and suggestions. In the version, we have reorganized the abstract by addressing the problem and proposed solution earlier. Otherwise, we mentioned that the challenges of large dataset management (in terms of storage and temporal performance) were published in our previous work [30]. This solution consisted on reducing the dimension of descriptors using PCA method that allowed to reduce the dimension with a rate of 70% with the maintain the same precision. This dimensional reduction was also applied for this work.
2. Question 2: What do you mean by combining SIFT and SURF? This should be discussed in one sentence how you combined them.
Response 2: the combination of SIFT and SURF descriptors consists of concatenating the extracted features (SIFT and SURF). In fact, the SIFT algorithm provides a matrix of n*m lines and 128 columns, while SURF provides a matrix of n*m lines and 64 columns, where n and m represent the image width and height. Since the two descriptors (SIFT and SURF) present a different number of columns, we have increased (using zero values) the size of SURF descriptor to 128 values in order to be compatible with the size of SIFT descriptor. Finally, we got one matrix with 2*(n*m) lines and 128 columns. The latter is used for indexing database images.
3. Question 3: the sentence “show the efficiency of our solution in terms of recall/precision” is not accurate. There is no relation between efficiency and precision or recall. These metrics measure the accuracy or performance.
Response 3: the new version is corrected. Indeed, the obtained results illustrated:
- Precision improvement in terms of recall/precision
- Performance improvement in terms of computation time as a result of exploiting GPUs
- Reduction of energy consumption.
4. Question 4: Please, provide a proper reference for (Markel, 2014) in line 51
Response 4: the reference [5] is corrected in the new version of the paper.
5. Question 5: you mentioned about three different retrieval categories (2D images, video, and 3D objects). If your method works with all these data, you need to describe how your method can work with all these data in the introduction
Response 5: in this paper, we were focused on the category of images only. However, the proposed solution of images retrieval can be applied for videos retrieval after detecting key frames as shown in our previous publication [6]. The 3D objects are not treated in this paper. We have corrected our mention in the new version of the paper.
6. Question 6: text in Fig. 1 and 2 hardly visible. You also may consider combining these figures by including the pre-processing step and images from figure 2 into figure 1.
Response 6: Fig. 1 and 2 have been corrected and fashioned in the new version of the paper.
7. Question 7: The title mentioned “Image Retrieval” but you present “Multimedia Retrieval” which are not consistent. CAD is not defined.
Response 7: we confirm that our proposal is for images retrieval. The new version is corrected mentioning images only. The CAD is the abbreviation of Computer Aided Diagnosis. This is corrected in the paper.
8. Question 8: make sure define all abbreviations in the manuscript. Please provide ref for the usage of image retrieval in different communities in the early introduction, for example, consider these works, shape-based retrieval, and analysis of 3D models, and, invariant feature descriptor based on harmonic image transform for plant leaf retrieval. Please consider using other feature-based methods such as, contour-based corner detection and classification by using mean projection transform, in line 26.
Response 8: The section of introduction has been improved and increased by the mention of use cases of images retrieval.
9. Question 9: are these categories “1. Multimedia content-based retrieval systems. 2. Cloud-based computer vision platforms are available in the literature or this is your finding? If you categorized this, you need to discuss why, if it is available in the literature, please include a proper citation.
Response 9: in fact, we can categorize two kinds of related works: Multimedia content-based retrieval systems. 2. Cloud-based computer vision platforms. This is not our finding; it represents the most significant work in the domain. The citations are corrected in the new version.
10. Question 10: Please consider discussing matching methods such as, invariant feature matching for image registration application based on new dissimilarity of spatial features, and, a region based fuzzy feature matching approach to content-based image retrieval, work in matching phase section.
Response 10: these methods and approaches have been included and discussed int the related work section
11. Question 11: why image features in figure 1 shows like a database? Moreover, the flow from the input image is from the bottom to the top, it is recommended to start from the top.
Response 11: in our case, we stored the features in a database in order get a fast access to the data using database management systems. The figure 1 is corrected accordingly to your remarks in the new version.
12. Question 12: text in Figure 3,4, and 4 hardly visible. Please include a reference (in the figure caption) to the image dataset used in Figure 3.
Response 12: the figures 3 and 4 have been corrected with the use of visible text. The two figures were designed by us, no reference is needed.
13. Question 13: a combination of SIFT and SURF mentioned in the abstract is not discussed properly in the method section.
Response 13: more details about the combination of SIFT and SURF descriptors mentioned in the new version (Section 2.1). Indeed, their combination consists of concatenating the extracted features (SIFT and SURF). In fact, the SIFT algorithm provides a matrix of n*m lines and 128 columns, while SURF provides a matrix of n*m lines and 64 columns, where n and m represent the image width and height. Since the two descriptors (SIFT and SURF) present a different number of columns, we have increased (using zero values) the size of SURF descriptor to 128 values in order to be compatible with the size of SIFT descriptor. Finally, we got one matrix with 2*(n*m) lines and 128 columns. The latter is used for indexing database images.
14. Question 14: you’ve got 100% precision and recall based on Figure 6(c) which is not realistic. Probably not enough data is used or you used a simple retrieval dataset. Moreover, it seems Wang dataset only contains 2D images while you mentioned about video and 3D objects earlier which is not consistent. You need to evaluate video and 3D objects, or alternatively, just talk about 2D image retrieval in the entire manuscript including the title.
Response 14: yes, we confirm that the precision of 100% is not realistic for all situations. Indeed, we obtained this score since the Wang database is not so big. However, we demonstrated that the efficient combination of descriptors allowed to improve significantly the precision compared the use of descriptors individually. To get more realistic results, we enriched our experiments (Section 4) in the new version using larger and more diversified databases such as Corel-10k and GIHIM. The Corel database presents 10000 images of size 192x128, classed in 100 categories where each class contains 100 images. The GHIM-10k database contains also 10000 images of size 300x400, classed in 20 classes where each category contains 500 images. These experiments confirmed our previous results by the improvement of accuracy thanks to the efficient combination of SIFT and SURF descriptors. The recall/precision were computing for the Top50 in addition to the Top20 similar images. Finally, in this work, we focus more of images indexation and retrieval.
15. Question 15: how many images did you use to produce Figure 6 results? Have you used all 10,000 + 1000 image in Wang dataset?
Response 15: the recall/precision values of Figure 6 were calculated using the Wang dataset with 10,000 images. As mentioned above, we applied experiment with bigger databases (GHIM-10k and Corel-10k). The evaluation was done for the Top50 most similar images. The experimental results show that our proposal allow to improve the precision. Moreover, our cloud-based solution provides an easier exploitation of our solution with a reduced time of execution thanks to the parallel exploitation of GPU’s computing units.

Reviewer 3 Report
1. In the abstract state and enhance the contributions. A large scale content based image retrieval platform that combines SIFT and SURF brings technological value but the original scientific aspect is not included.
2. The overall paper structure is good but it lacks the scientific contributions. An image retrieval system based on SIFT and SURF is not so new.
Author Response
a. Reviewer 3:
1. Question 1: in the abstract state and enhance the contributions. A large-scale content-based image retrieval platform that combines SIFT and SURF brings technological value, but the original scientific aspect is not included.
Response 1: thank you for these interesting remarks. In fact, our research consisted on selecting the best descriptors for images retrieval and that must be adaptable for dimensionality reduction. On the one hand, we proposed a combination of descriptors that allows to improve the precision. On the other hand, the proposed combination of descriptors is well adapted for dimensionality reduction, where the selection of most significant values of descriptors (using PCA method) allowed to reduce the research time with the maintain of precision [30]. As result, our method is well suited for large scale images.
2. Question 2: the overall paper structure is good, but it lacks the scientific contributions. An image retrieval system based on SIFT and SURF is not so new.
Response 2: in fact, the use of SIFT and SURF in not so new. Our contribution consists of exploiting both of them in a coordinate way after image denoising. On the other hand, our contribution is presented within three parts:
a. the development of an efficient method of content-based image retrieval that combines the descriptors of SIFT and SURF
b. a portable GPU implementation that allows to accelerate the process of indexation and research within multimedia databases. This implementation allows to exploit both NIVIDIA and AMD/ATI cards.
c. cloud-based implementation that allows an easier exploitation of our GPU-based method without the need to download, install and configure software and hardware. The platform handles multi-user connection based on Docker containers orchestration architecture.
Finally, the new version of the paper has been improved by validating results with larger and more diversified databases such as Corel-10k and GIHIM. The Corel database presents 10000 images of size 192x128, classed in 100 categories where each class contains 100 images. The GHIM-10k database contains also 10000 images of size 300x400, classed in 20 classes where each category contains 500 images. These experiments confirmed our previous results by the improvement of accuracy thanks to the efficient combination of SIFT and SURF descriptors. The recall/precision were computing for the Top50 in addition to the Top20 similar images.

Round 2
Reviewer 1 Report
Most of the issues have been modified. The current manuscript could be accepted.
Author Response
Thank you for the interesting remarks that alowed to improve the quality of the paper.
Reviewer 2 Report
All the comments have been addressed. Please consider bellow minor changes:
The accuracy in Table 1 should be a percentage, I'm not sure what kind of scaling you used. Please double check.
Extend Figure 2 caption and add more information about each section of the Figure.
Start Introduction with Numbering starting from 1, not 0. Also, fix numbering for the subsequence sections.
Author Response
Thank you for the interesting remarks that have been well taken into account ine the newe version of the paper. Figure 1 and sections numbers are well corrected.
For the accuracy in Table 1, we presented the speedup (accelaration) rates. This sppedup was corrected also in the new version.
Reviewer 3 Report
The paper has been improved based on previous remarks. Author's response provide sufficient detail to clarify previous issues.
Author Response
Thank you for the interesting remarks that were so useful for improving the quality of the paper.